# Field programmable spin arrays for scalable quantum repeaters

Hanfeng Wang[1,2], Matthew E. Trusheim ®[1,3] ✉, Laura Kim[1,4], Hamza Raniwala[1,2] & Dirk R. Englund ®[1,2] ✉

The large scale control over thousands of quantum emitters desired by quantum network technology is limited by the power consumption and cross-talk inherent in current microwave techniques. Here we propose a quantum repeater architecture based on densely-packed diamond color centers (CCs) in a programmable electrode array, with quantum gates driven by electric or strain fields. This 'field programmable spin array' (FPSA) enables high-speed spin control of individual CCs with low cross-talk and power dissipation. Integrated in a slow-light waveguide for efficient optical coupling, the FPSA serves as a quantum interface for optically-mediated entanglement. We evaluate the performance of the FPSA architecture in comparison to a routing-tree design and show an increased entanglement generation rate scaling into the thousand-qubit regime. Our results enable high fidelity control of dense quantum emitter arrays for scalable networking.

Future quantum repeaters or modular quantum computers will need to manage large numbers of multiplexed memory qubits with efficient local operations. Solid-state artificial atoms such as color centers (CCs) in diamond are promising quantum memories[1,2]. Precision control of the electronic spin ground state of CCs presently relies on AC magnetic fields[3–6]. Developing architectures for spatially multiplexed microwave control with sufficiently low power dissipation and cross-talk remains an open challenge. Previous work achieved localized control using a magnetic field with a spatial gradient[7], or by producing a spatially-varying detuning of the CC resonant frequency using a gradient magnetic[8] or optical[9] field combined with global magnetic addressing. Here, we propose a fundamentally different approach that uses highly localized fields−either strain or electric, depending on the CC of choice−which can be driven capacitively for low power dissipation. Electric field-based spin control has been proposed in several systems[10–12], while strain driving has been demonstrated for many CC systems[13,14]. We show that these approaches offer lower power dissipation and cross-talk, as well as compatibility with integrated circuit (IC) platforms likely needed for scaling. With an efficient optical interface and all-to-all connectivity, our platform can be integrated to allow scalable entanglement generation.

We consider a programmable array of electrodes positioned around arrays of CCs in a diamond waveguide. This 'field programmable spin array' (FPSA) architecture has three key elements: (1) A quantum memory; for the electric-driving case the diamond nitrogen-vacancy (NV) center, which has already been used for optical entanglement distribution across as many as three qubits[15–17], and for strain driving the diamond silicon-vacancy (SiV) center, which has been used to demonstrate memory-enhanced quantum communication[18]. (2) An efficient optical interface through a slow-light photonic crystal (PhC) waveguide enabling ~ 25 × Purcell enhancement of the CC's coherent transition. Compared to cavity structures, the slow-light waveguide can host a large number of qubits and still maintain relatively high Purcell enhancement. 3. An electrode array positioned along individual qubits in the waveguide. We estimate that the FPSA enables ~ 100 ns-duration spin rotations, as well as ~ 600 GHz-range DC tuning of CC optical transitions.

The article is organized as follows. We first introduce the FPSA and estimates achievable gate performance, focusing chiefly on duration and cross-talk. Then co-designed PhC waveguide is described to achieve high emitter-waveguide coupling with low optical loss, as well as dynamical optical tunability. The final section combines the

[1]Research Laboratory of Electronics, M.I.T., 50 Vassar Street, Cambridge, MA 02139, USA. [2]Department of Electrical Engineering and Computer Science, M.I.T., Cambridge, MA 02139, USA. [3]DEVCOM, Army Research Laboratory, Adelphi, MD 20783, USA. [4]Department of Materials Science and Engineering, University of California, Los Angeles, Los Angeles, CA 90095, USA. ✉e-mail: mtrush@mit.edu; englund@mit.edu

elements of two previous sections to show a quantum repeater architecture enabled by the FPSA, illustrating how the FPSA can mediate local qubit interactions and multiplexed quantum network connectivity. In the first several sections we first explore the case of NV centers in diamond, driven by electric fields, and second the inversion-symmetric group IV emitters controlled via strain fields. In the final section we consider NV centers as the exemplary CC for quantum architecture design.

## Results

### Localized single-qubit control
Figure 1 shows the FPSA design. It consists of a single-mode diamond waveguide hosting a centered CC array, placed onto dielectric fins between an electrode array. For the electric-driving approach, we use HfO$_2$ as it has a high dielectric constant of 23 in the radio frequency range and a relatively low index of 1.9 in the optical range[19,20]. This allows it to concentrate the low frequency electric field required for spin coupling while guiding the optical mode. In the case of strain driving, we use piezoelectric AlN fins that produce the strain field while also producing a periodic modulation of the refractive index ($n = 2.16$). We assume conductive indium tin oxide (ITO) as the electrode material to minimize optical loss. Finally we use SiO$_2$ as a low-index substrate.

As illustrated in Fig. 1, we consider an array of CC spin memories approximately at a periodic spacing $a$, i.e., the $k$th CC has a position $\vec{r}_k = ka\hat{z} + \vec{\delta}_k$, where $|\vec{\delta}_k|/a \ll 1$.

### Electric field-driven quantum gates
We first consider NV centers as an exemplary CC, and show that electric fields can produce high-fidelity localized quantum gates. The relevant interaction between an NV ground state spin and an electromagnetic field $\vec{E}$ and $\vec{B}$ is captured in the Hamiltonian[14,21,22]:

$$H_{\vec{E},\vec{B}}/h = \gamma_S \vec{S} \cdot \vec{B} + d'_\perp [\{S_{x'}, S_{z'}\}E_{x'} + \{S_{y'}, S_{z'}\}E_{y'}] \\ + d_\parallel S_{z'}^2 E_{z'} + d_\perp [(S_{y'}^2 - S_{x'}^2)E_{x'} + \{S_{x'}, S_{y'}\}E_{y'}] \tag{1}$$

where $d_\perp = |\hat{z}' \times (\hat{z}' \times \vec{d})| = 17$ Hz cm/V ($d_\parallel = \hat{z}' \cdot \vec{d} = 0.35$ Hz cm/V) denotes the perpendicular (parallel) part of spin-electric field

susceptibility[21], $h$ the Planck constant, $\mathbf{S}$ the electron spin operator, and $\gamma_S$ the gyromagnetic ratio. $d'_\perp$ has not been quantified experimentally or theoretically, but is estimated near 1/50 $d_\perp$[14,23]. Here we use the primed coordinates ($x', y', z'$) to indicate the coordinates relative to the NV aligned along the $z'$ axis.

We now consider an external electric driving field $\vec{E}_{\pm 1 \leftrightarrow 0}$ ($\vec{E}_{+1 \leftrightarrow -1}$) resonant with the $|\pm 1\rangle \leftrightarrow |0\rangle$ ($|+1\rangle \leftrightarrow |-1\rangle$) transitions, which are non-degenerate under a small bias magnetic field along the $z'$-axis, as shown in Fig. 1a. From the Schrödinger equation, the Rabi frequency of coherent driving on the NV ground state triplet is:

$$h\Omega_R^{\pm 1 \leftrightarrow 0} = \frac{1}{\sqrt{2}} d'_\perp |\vec{E}_\perp(\vec{r}_k)| \\ h\Omega_R^{+1 \leftrightarrow -1} = d_\perp |\vec{E}_\perp(\vec{r}_k)| \tag{2}$$

where $\vec{r}_k$ indicates the positions of NVs shown in Fig. 1 and $\vec{E}_\perp(\vec{r}_k)$ is the component of electric field perpendicular to NV axis. As shown in Fig. 1a, b, we choose $\vec{\mu}_1 = [1\bar{1}0]$ and $\vec{\mu}_2 = [\bar{1}\bar{1}2]$ as basis vectors for the plane perpendicular to NV axis, i.e. $\vec{E}_\perp(\vec{r}_k) = (E_{\vec{\mu}_1}(\vec{r}_k), E_{\vec{\mu}_2}(\vec{r}_k))$ where $\vec{\mu}_1$ and $\vec{\mu}_2$ are also the axes of NV optical transitions $E_y^{op}$ and $E_x^{op}$, respectively[24].

Absent other experimental noise, the single-qubit gate fidelity is limited by the inhomogeneous dephasing time $T_2^* \sim 10 \mu s$[25]. For a pure superposition state, the fidelity of a $\pi$-rotation at Rabi frequency $\Omega_R$ under this dephasing process is given by $F_{\text{dephasing}} = 1/2(1 + \exp(-1/2\Omega_R T_2^*))$. Considering random pure states uniformly-distributed on the Bloch sphere, the average fidelity reaches above 0.99 with a Rabi frequency of 1.7 MHz. For double quantum transition, a resonant electric field of 10 V/μm is needed to reach this gate fidelity. In our geometry, this requirement is met for a ~10 V potential difference, which is compatible with modern integrated circuits technology such as complementary metal-oxide semiconductor (CMOS) platforms. We estimate that electric field-driven Rabi frequencies can reach $\Omega_R^{+1 \leftrightarrow -1} \sim 0.13$ GHz and $\Omega_R^{\pm 1 \leftrightarrow 0} \sim 1.9$ MHz, limited by diamond's dielectric strength $E_{bd}^{dmd} \sim 2 \times 10^3$ V/μm[26,27] and HfO$_2$'s dielectric strength $E_{bd}^{HfO_2} \sim 1.6 \times 10^3$ V/μm[28] at a separation of hundreds of nm.

The electric field profile of an example FPSA is shown in Fig. 2a, which plots the $|\vec{E}_\perp(x,y,z=0)|$ electric field component obtained from

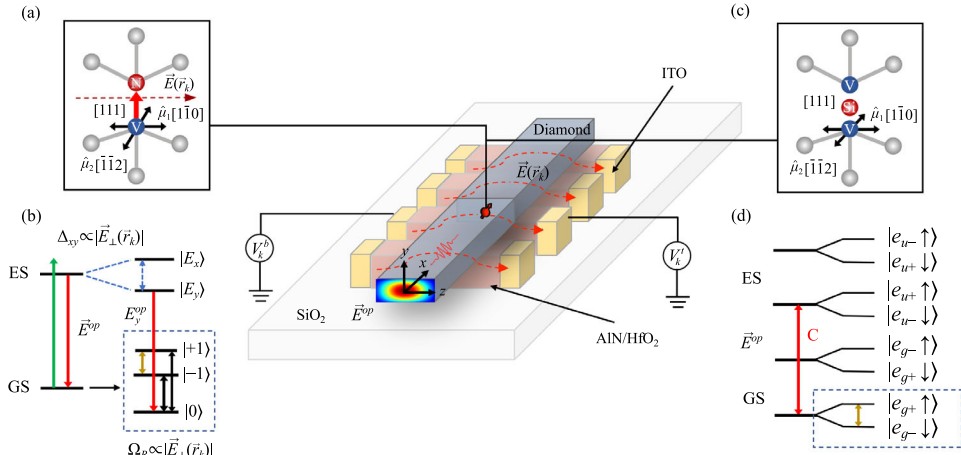

**Fig. 1 | An exemplary FPSA implementation. A diamond waveguide (gray) containing CCs is positioned on an array of high-index dielectric fins (red) between ITO electrodes (yellow) connected to a voltage supply.** Electric fields $\vec{E}(\vec{r}_k)$ are generated by top (bottom) contacts at potentials $V_k^t(V_k^b)$ for spin control. The CC fluorescence couples to the waveguide mode $\vec{E}_z^{wg}$. Here the $\hat{x} = [001], \hat{y} = [110], \hat{z} = [1\bar{1}0]$. **a** The NV center in diamond in the [111] direction with transition dipoles $\vec{\mu}_1 = [1\bar{1}0]$ and $\vec{\mu}_2 = [\bar{1}\bar{1}2]$. The FPSA field is closely aligned to the $\vec{\mu}_1$ transition, and perpendicular to the spin quantization axis. **b** A Zeeman shift in an external magnetic field splits the NV spin $m_s = \pm 1$ sublevels, while an electric field splits the excited state levels $|E_x\rangle$ and $|E_y\rangle$ by $\Delta_{xy}$ via the DC Stark effect (left). The black (yellow) line indicates the Rabi transition between $|\pm 1\rangle \leftrightarrow |0\rangle$ ($|+1\rangle \leftrightarrow |-1\rangle$) driven by a resonant transverse electric field. **c** The SiV center in diamond in the [111] direction with transition dipoles $\vec{\mu}_1 = [1\bar{1}0]$ and $\vec{\mu}_2 = [\bar{1}\bar{1}2]$. **d** Level structure of SiV. The optical `C' transition is resonantly addressed, and enhanced by the FPSA slow-light mode. The $|e_{g+} \uparrow\rangle$ and $|e_{g-} \downarrow\rangle$ levels are split in an external magnetic field, and used as a qubit with strain-driven quantum control.

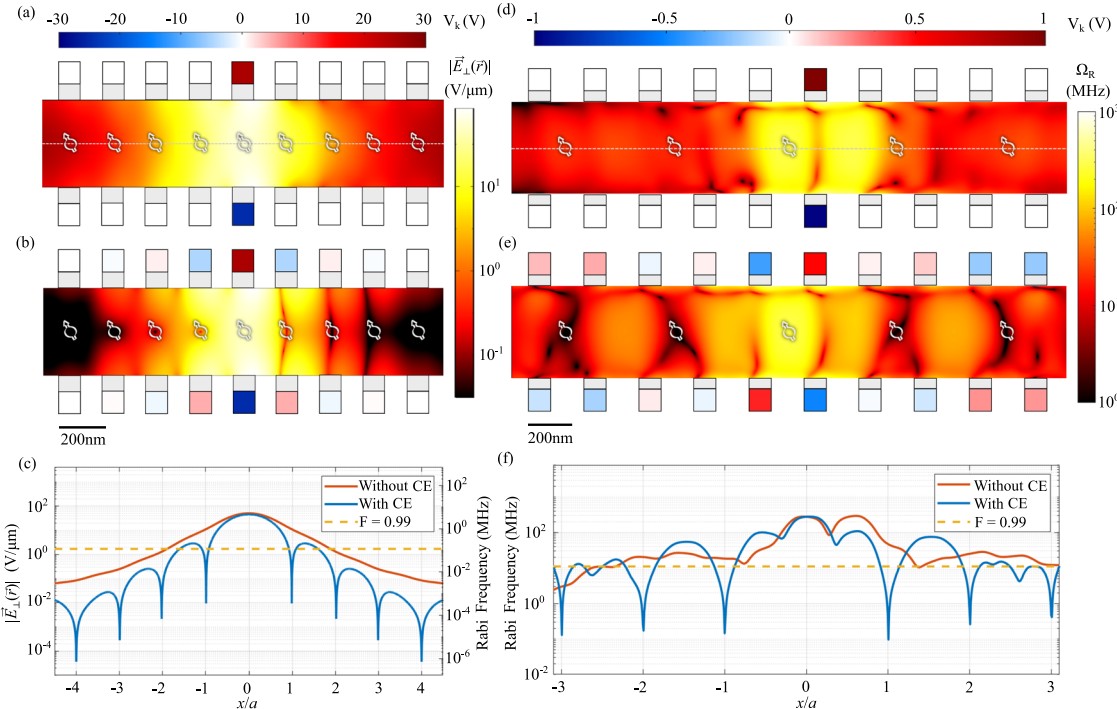

**Fig. 2 | Electric and strain field profiles. a** Electric field component $|\vec{E}_\perp(x,y=0,z)|$ for a applied voltage $V_k$ on a single pair of electrodes. **b** $|\vec{E}_\perp(x,y=0,z)|$ for voltage settings optimized for reduced cross-talk. **c** Electric field component $|\vec{E}_\perp(x,y=z=0)|$ along central axis of the FPSA (dotted line in **a**) for the voltages configurations with and without crosstalk elimination (CE), and corresponding Rabi frequency of the $|+1\rangle \leftrightarrow |-1\rangle$ transition. **d** Rabi frequency between $|e_{g+}\uparrow\rangle \leftrightarrow |e_{g-}\downarrow\rangle$ for a single electrode pair in the strain-driving geometry. **e** Rabi frequency between $|e_{g+}\uparrow\rangle \leftrightarrow |e_{g-}\downarrow\rangle$ after cross-talk elimination. **f** Rabi frequency along the white dotted line in **d**.

Maxwell's equations using COMSOL Multiphysics when a voltage $V_k = V_k^t - V_k^b = 50$ V is applied in FPSA with parameters in Table 1.

## Strain-driven quantum gates

Although NV is widely investigated, Group-IV centers in diamond have drawn interest due to their inversion symmetry and optical properties[29]. However, due to the inversion-symmetry, the spin and orbital transitions of Group-IV emitters are almost immune to electric fields[30]. We then propose to use a strain field for quantum control. The Rabi frequency when an oscillating strain field is resonant with the $|e_{g+}\uparrow\rangle \leftrightarrow |e_{g-}\downarrow\rangle$ transition is[13,31]:

$$\Omega = \frac{\gamma_S \sqrt{\beta^2 + \gamma^2} B_\perp}{\lambda_{SO}} \tag{3}$$

where $\beta$ and $\gamma$ are the magnitudes of transverse AC strain fields that couple to the SiV spin, $B_\perp$ is a static transverse magnetic bias field, and $\lambda_{SO}$ is the spin-orbit coupling strength. We consider a resonant strain field generated in an FPSA structure as shown in Fig. 1, where the electric field produces a strain field in piezoelectric AlN fins rather than being guided using HfO$_2$ as previous. The Rabi frequency induced by a strain field in the FPSA geometry is shown in Fig. 2d when a voltage

$V_k = V_k^t - V_k^b = 2$ V is applied, assuming a transverse bias field of $B_\perp = 0.17$ T as used in prior work[31]. The parameters of the device are listed in Table 1. Note that the electrodes are placed off-center from the emitter, with four total electrodes per unit cell. These added controls allow for manipulation over the additional degrees of freedom of strain fields as compared to electric fields (see Supplementary). To achieve a $\pi$ rotation of the spin degree of freedom with $F = 0.99$, a ~0.01 V potential difference is needed in our structure—within the range of modern integrated circuit technology such as CMOS platforms.

## Control cross-talk

The closest separation between individually controllable CCs in an array is limited by the cross-talk between the target CC at location $\vec{r}_k$ and its nearest-neighbor CC at $\vec{r}_{k+1}$. During a $\pi$-pulse on qubit $\vec{r}_k$, there is an undesired rotation on $\vec{r}_{k+1}$. We evaluate the cross-talk fidelity $F_C$ by comparing $\mathbf{R}(\vec{r}_{k+1})$ and desired identity operation $\mathbf{I}(\vec{r}_{k+1})$. This fidelity can be expressed as

$$F_C(\mathbf{R}(\vec{r}_{k+1}), \mathbf{I}(\vec{r}_{k+1})) = \left( \text{tr} \sqrt{\sqrt{\rho}\sigma\sqrt{\rho}} \right)^2 \tag{4}$$

where

$$\begin{aligned} \rho &= \mathbf{R}(\vec{r}_{k+1}) |\psi_0(\vec{r}_{k+1})\rangle\langle\psi_0(\vec{r}_{k+1})| \mathbf{R}^\dagger(\vec{r}_{k+1}) \\ \sigma &= \mathbf{I}(\vec{r}_{k+1}) |\psi_0(\vec{r}_{k+1})\rangle\langle\psi_0(\vec{r}_{k+1})| \mathbf{I}^\dagger(\vec{r}_{k+1}) \end{aligned} \tag{5}$$

and $|\psi_0(\vec{r}_{k+1})\rangle$ is the initial quantum state of CC at location $\vec{r}_{k+1}$. For the profile shown in Fig. 2a, $F_C^{fin} = 0.92$. The field confinement provided by the HfO$_2$ fins results in a significant improvement over bare electrodes, where $F_C^{bare} = 0.69$ (see Supplementary).

For localized strain driving structures shown in Fig. 2d, we calculate a cross-talk fidelity of 0.88. The cross-talk fidelity for strain

## Table 1 | FPSA Parameters

| | Symbol | Value (Electric) | Value (Strain) |
|---|---|---|---|
| Spin memory spacing | $a$ | 183 nm | 488 nm |
| Diamond $y$-dimension | $h_{wg}$ | 91 nm | 91 nm |
| Diamond $z$-dimension | $w_{wg}$ | 364 nm | 410 nm |
| HfO$_2$/AlN $x$-dimension | $l_{fin}$ | 91 nm | 98 nm |
| HfO$_2$/AlN $y$-dimension | $h_{fin}$ | 273 nm | 273 nm |
| HfO$_2$/AlN $z$-dimension | $w_{fin}$ | 500 nm | 500 nm |

tuning is lower than that of the electric field case because the acoustic wavelength (~μm) is comparable with the device size. A propagating acoustic wave is launched along the waveguide, which makes it difficult to localize the strain field compared to the electric field.

## Cross-talk elimination

To further reduce the cross-talk, we use our individual control over $2N$ voltages $V = \{V_k^{t(b)}\}$ to eliminate the driving field at the locations of the non-target qubits. Electric field applied on each qubit $\vec{E}_\perp = \{E_{\vec{\mu}_1}(\vec{r}_k), E_{\vec{\mu}_2}(\vec{r}_k)\}$ has a linear dependence with the voltage by $\vec{E}_\perp = GV$, where $G_{ij}$ is the linear map between $E_i$ and $V_j$, computed by COMSOL Multiphysics. $V$ are then chosen to minimize the cross-talk by $V_{CE} = G^{-1}\vec{E}_{tar}$. For the case of a single-qubit gate on a target NV at location $i$, we set $\vec{E}_{tar} = \vec{E}_{\perp,tar}^T \otimes_K \delta_{ik}$ and $\vec{E}_{\perp,tar}$ is the AC electric field applied on the target NV. Since the number of independent degrees of freedom is equal to the number of electric field values to be minimized, this inversion is possible. As shown in Fig. 2b, c, the cross-talk elimination process creates low electric field on non-target NV positions, increasing the cross-talk fidelity to $F_{CE} > 0.99$. The tolerance of the NV position to achieve this fidelity is 18 nm (73 nm) for the first (second) nearest neighbors shown in Fig. 2c. Now, the total infidelity is mainly caused by dephasing rather than cross-talk. This procedure can be used for any arbitrary operations over all NVs by the above procedure, choosing a specific $\vec{E}_{\perp,tar}$.

Although the strain field is a tensor, a similar cross-talk elimination process can be applied for strain tuning, as shown in Fig. 2e and Fig. 2f. the cross-talk fidelity can go to $F = 0.99$ with a tolerance of NV position of ~110 nm along $x$-axis. The distance along $x$-axis needed to achieve cross-talk fidelity $F = 0.99$ is 1.2 μm (0.36 μm) for strain (electric) field, meaning electric field can be more localized in FPSA design.

## Heat load for electric field- vs magnetic-field-based spin control

Heat dissipation is critically important in cryogenic environments, where cooling power is limited and heating can degrade performance. We approximate the low-temperature stage power consumption of electric field-based coherent control by modeling the FPSA as a capacitance $C$ with a parallel resistance $R$ in series with a wire (resistance $R_w \sim 10^{-2}\,\Omega$) inside the cryogenic environment (circuit details in Supplementary). We choose the figure of merit as the energy per spin π-pulse that is deposited at the low-temperature stage. In our design, the FPSA acts as an open circuit, and almost all the power is reflected back to the high temperature region. For $|+1\rangle \leftrightarrow |-1\rangle$ transition, the energy dissipation in the cryostat is given by:

$$J_E = \frac{1 + \omega^2 C^2 R_w R}{R} \frac{\Lambda^2 \Omega_R}{4 d_\perp^2} \tag{6}$$

where $\omega$ is the frequency of the AC electric field, and $\Lambda \sim 1\,\mu m$ is the characteristic length that relates the applied voltage on the FPSA and electric field at the positions of NVs. In our geometry, $C = 4.9 \times 10^{-17}$ F is the circuit capacitance simulated by COMSOL and $R \sim 10^{20}\,\Omega$ is the resistance of the HfO$_2$ calculated from thin film resistivity[32]. The energy dissipation in the cryostat per π-pulse $J_E$ for a Rabi frequency $\Omega_R = 1.7$ MHz is $1.4 \times 10^{-21}$ J with a peak maximum current $I = 98$ mA. A second figure of merit is the dissipation ratio between electric field and magnetic field-based driving with the same Rabi frequency, $J_E/J_B$. Here we take the magnetic circuit to be the bare wire with a resistance $R_w$ and a capacitance $C_w$, with the NV positioned at a distance $\Lambda$ from the wire. Then the ratio is given by:

$$\frac{J_E}{J_B} \sim \frac{\mu_0^2 \gamma^2}{4\pi^2 d_\perp^2} \frac{1 + \omega^2 C^2 R_w R}{R_w R} \tag{7}$$

where $\mu_0$ is the vacuum permeability. Here $J_E/J_B = 1.0 \times 10^{-5}$, suggesting the power dissipation for electric field control is 6 orders of magnitude less than that for magnetic field control. For driving a single quantum transition $|0\rangle \leftrightarrow |\pm1\rangle$ $J_E/J_B = 5.0 \times 10^{-2}$. In the real case, we need to consider the leakage current. For a ~ PΩ leakage resistance[33], we will have $J_E/J_B = 1.2 \times 10^{-4}$ $(6.0 \times 10^{-1})$ for $|+1\rangle \leftrightarrow |-1\rangle$ $(|\pm1\rangle \leftrightarrow |0\rangle)$ transitions. A similar calculation can be made for strain-based FPSA. Assuming a transverse bias field of $B_\perp = 0.17$ T as used in prior work[31] with $U_e = 0.14$ V to reach $\Omega_R = 20$ MHz and a ~ PΩ leakage resistance[33], the heat load per π-pulse in the low-temperature part for this circuit is $2.5 \times 10^{-25}$ J. For a comparison with microwave control, we assume $\beta = -1.29 \times 10^2$ GHz based on[34] and the heat load ratio $J_S/J_B = 8.9 \times 10^{-9}$. See Supplementary Note 6 for details. Here we only consider the heat load in low temperature part. For high temperature part, the heat generated on the transmission line for strain-based FPSA is $2.5 \times 10^{-12}$ J with a peak current of 1.4 mA, which is much larger than the heat load in cryogenic stage.

## Efficient coupling of an NV array to a slow-light PhC waveguide

The entanglement rate of NV centers relies on the spin-photon coupling efficiency, which is given by[35,36]:

$$\beta = \frac{F_P \cdot \Gamma_{wg0}}{F_P \cdot \Gamma_{wg0} + \Gamma_{others}} \tag{8}$$

where $\Gamma_{wg0}$ is the decay rate of spin-entangled transition in the absence of any optical structures, and $\Gamma_{others}$ the total rate of all other decay mechanisms. Slow-light waveguide structures produce a photonic bandgap, resulting in a small group velocity near the band edge. As a result, an emitter placed in the mode maximum of a slow-light waveguide experiences a large enhancement in the local density of electromagnetic states, and its rate of transition is enhanced by the Purcell factor, $F_p$[36,37].

Conveniently, the fin structures provide a periodic dielectric perturbation, forming a slow-light mode in the optical band. Fig. 3a indicates the TE-like modes of the slow-light waveguide with the parameters shown in Table 1. Here we focus on the $E_y^{op}$ transition of the NV center with a frequency $\nu_0$. By coupling the NV transition at $\nu_0$ to the slow-light region, we can thus funnel the coherent emission into waveguide modes near wave-vector $k_x(\nu_0)$, as shown in Fig. 3a. From finite difference time domain (FDTD) simulations (Lumerical), we obtain a maximum Purcell factor of $F_{P\max} = 25$ for an NV in the [111] direction placed on the mid plane of the diamond waveguide when the number of periods is 100 (Fig. 3b).

The total photon collection efficiency out of the waveguide is given by $\eta_{wg} = \beta \exp(-t_{wg}N)$, where $t_{wg}$ is the waveguide transmission from the emitter to the waveguide facet. Assuming a NV Debye-Waller factor of $DW = 0.03$[23] and we use the relation $\Gamma_{others}/\Gamma_{total} = 1 - DW$ to calculate $\beta = 25\%$[38,39]. The transmission loss $t_{wg} \sim 8 \times 10^{-4}$ dB/period is estimated from FDTD simulations.

## Spectral addressing by localized optical tuning

To selectively couple the waveguide propagating modes to a specific CC center in the array, we use the electrodes for another function: to tune the optical transition frequency of individual CCs. In the case of NVs under electric field control, the emitter's natural $E_y^{op}$ transition at $\nu_0$ is shifted to $\nu_0 + \Delta\nu_0$, where $\Delta\nu_0$ is given by[40]:

$$h\Delta\nu_0 = \Delta\mu_\parallel E_\parallel - \frac{\sqrt{2}}{2}\mu_\perp \sqrt{E_{\vec{\mu}_1}^2 + E_{\vec{\mu}_2}^2} \tag{9}$$

where $E_\parallel$ is the electric field along [111], $\Delta\mu_\parallel = \mu_\parallel^{GS} - \mu_\parallel^{ES} \sim 1.5$ D is the parallel dipole moment difference between excited states and ground states, and $\mu_\perp \sim 2.1$ D is the perpendicular component of electric dipole

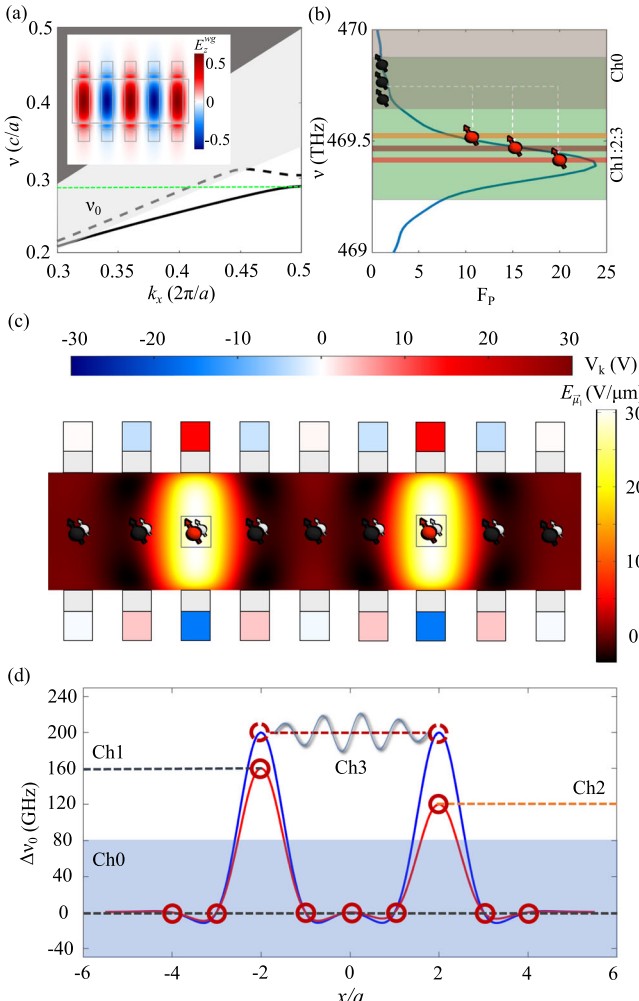

**Fig. 3 | Coupling of quantum emitters to slow-light waveguide with spectral addressing. a** Photonic TE band structure of the FPSA slow-light waveguide using the parameters shown in Table 1. The dark (light) shaded regions indicate the light cone for propagation in free space (substrate). The inset shows the $z$-component of the electric field at the midplane of the diamond. **b** Purcell factor for the FPSA with a finite number of periods $N = 100$ near the bandgap. Green shading indicates the Stark tuning range, and the horizontal shading indicates non-interacting frequency channels. **c** Electric field profile for placing two NVs at $j = \pm 2$ on resonance. **d** NV Stark shift versus positions for two voltage configurations. The blue curve shows the Stark shift for the electric field profile shown in **c**. The red curve shows another voltage setting where two emitters are tuned to non-interacting channels.

moment. Here we choose $E_y^{op}$ transition to avoid the depopulation and mixing of excited states at large applied fields[41].

The maximum tuning range using this effect is ~600 GHz assuming an applied electric field of $10^2$ V/μm in the FPSA architecture, indicated by the green shaded area in Fig. 3b. This corresponds to tuning across the full range of the slow-light Purcell enhancement and into the waveguide bandgap. As the Purcell-enhanced NV ZPL transition linewidth is ~100 MHz, the wide range of the Stark tuning allows multiple-frequency channels in which NVs can be individually addressed in the frequency domain. Three channels (Ch1 – 3) spaced by 40 GHz and an off-resonant channel (Ch0) are indicated in Fig. 3b. The Purcell enhancement across Ch1 – 3 is maintained at ~10, while the large spacing suppresses interactions (e.g. photon absorption) between channels[42]. An analogous effect can be achieved using strain tuning[31], see Supplementary Note 6 for details.

The ability for FPSA to reconfigure the electric field profile locally allows for arbitrary and independent configuration of NV optical

transitions. Unlike in Sec. I where the $E_\parallel$ is neglected, here we need $3N$ degree of freedoms to control components $E_\parallel$, $E_{\vec{\mu}_1}$ and $E_{\vec{\mu}_2}$ for each of the $N$ NVs. However, we can use symmetry $V_t + V_b = 0$ to set $E_\parallel = E_{\vec{\mu}_2} = 0$ in the ideal case. The remaining $N$ degrees of freedom can be used to set $E_{\vec{\mu}_1}$ for all the NVs. For example, we compare two configurations in Fig. 3d. The blue curve shows the frequency shift for an electric field profile in Fig. 3c, resulting in two NVs on resonance. The red curve shows the frequency shift for a different voltage setting, where two NVs are in different channels without interaction. In both cases, all other NVs are off-resonant in Ch0. The ability to dynamically control the NV transition frequency via electrical control can then be used to perform individual emitter initialization and readout, and to reconfigure quantum network connectivity as described below.

## Quantum repeater performance

We next consider the application of the FPSA as a quantum repeater to generate Bell pairs $|\psi_{AB}\rangle$ between two memory qubits at Alice and Bob (A and B), as illustrated in Fig. 4a. Many quantum repeater protocols have been proposed in previous works, including the use of quantum emitters as quantum memories[43], and multiplexing schemes that achieve all-to-all connectivity[44,45]. Here we show the FPSA can improve the repeater performance via an improved scaling with number of qubits. The repeater protocol has two steps:

(1) Distant entanglement between A(B) and electron spin $|j_e\rangle (|k_e\rangle)$ in the FPSA. Many schemes are proposed for entanglement generation within cavities and waveguides[46–48]. Here we choose a heralded single-photon scheme, which has previously been demonstrated for NV centers[16], followed by swapping to the $^{15}N$ nuclear spin $|j_n\rangle (|k_n\rangle)$. For each link, we assume a length-$L$ noiseless channel with transmission $\eta = \exp(-\gamma L)$, where $\gamma = 0.041$ km$^{-1}$[44]. Each entanglement attempt has a success probability of $p_1 = 2\alpha\eta p_d p_c \eta_{wg}/2$, where $p_d = 0.83$ ($p_c = 0.33$) is the detection (quantum frequency conversion, if necessary) efficiency[49,50]. We conservatively assume a lower $F_P = 10$ ($\beta = 25\%$) to avoid high loss and fabrication sensitivity in the regime of high group index[51]. Here we set $\alpha = 0.01$ to keep the two-photon excitation error of this scheme below 1%[16].

(2) Local entanglement swapping to generate entanglement between A and B. First, (2.i) CC electron spins $|j_e\rangle$ and $|k_e\rangle$ are Stark-shifted to the same optical transition and entangled via the two-photon Barrett-Kok scheme[17,48] with success probability $p_2 = (p_d \eta_{wg})^2/2$. Then, (2.ii) A CNOT gate is performed between electron spin and nuclear spin to establish the local entanglement, then Bell measurements in the electron-nuclear spin basis of the memories $j$ and $k$ swap the local entanglement to distant entanglement of A and B after subsequent feed-forward[52,53]. This step makes use of the FPSA's all-to-all connectivity to realize a 'quantum router' architecture[44] that minimizes the latency (waiting time and associated decoherence) and local buffer size.

To evaluate the performance, we consider the entanglement rate $\Gamma_{AB}$ as the figure of merit, defined as the average number of generated Bell pairs $|\psi_{AB}\rangle$ per second. For two qubits, $\Gamma_{AB}$ is the inverse of total time used for single pair entanglement generation. Parallel operations of $N$ pairs can increase this rate by a factor of $N$. The FPSA time-multiplexes spin-photon entanglement to A and B, sending spin-entangled photons from different emitters in short succession. It is implemented by Stark-shifting the optical transition of selected color centers $j$ and $k$ from Ch0 to Ch1 and Ch2 (shown in Fig. 4a), while all other 'unselected' color centers remain in Ch0. After an entanglement generation attempt, the NVs $j, k$ are tuned back into Ch0 and the process is repeated with the subsequent pair of NVs $j, k = j + 1, k + 1$ as shown in Fig. 4a. In this way, time-multiplexing channels allow $N_{ch} = t_{link}/t_{ph}$ qubits operate in parallel, where $t_{link}$ is the heralding time for an optical pulse traveling photon in the fiber link and $t_{ph}$ is the CC

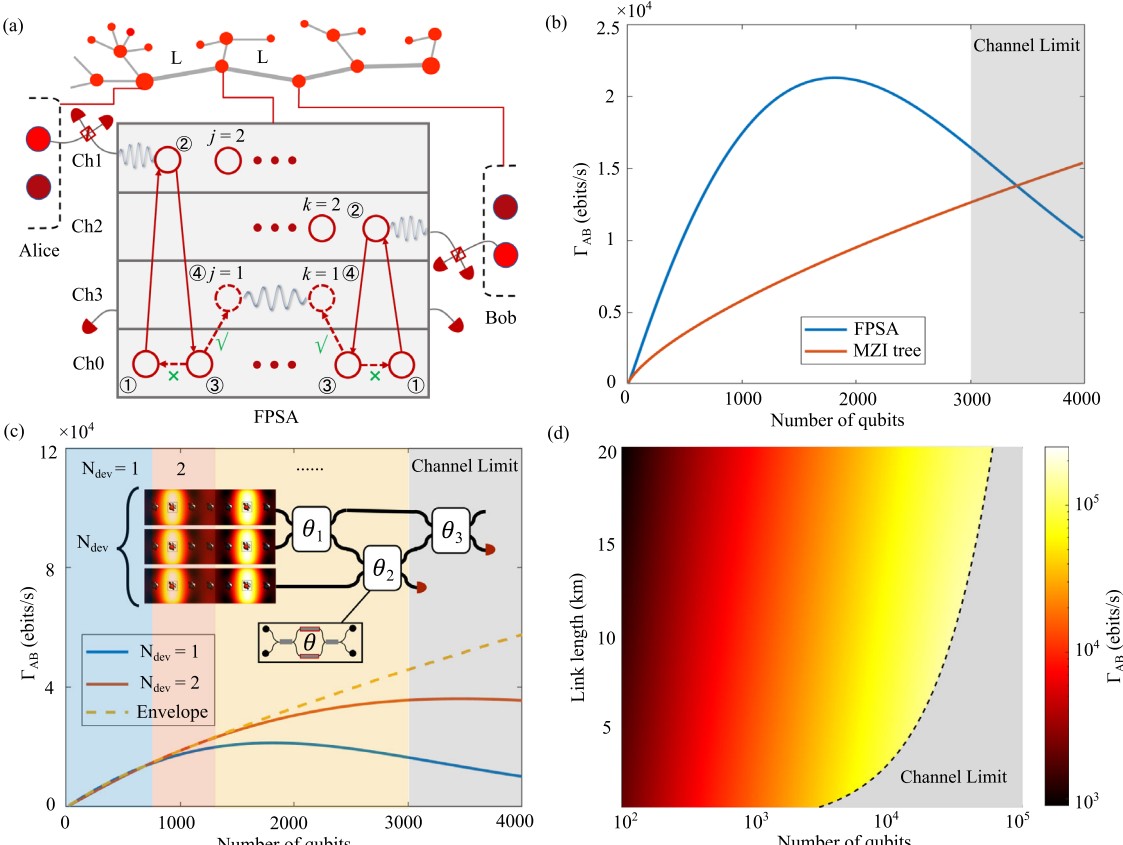

**Fig. 4 | Quantum repeater performance. a** Quantum repeater architecture containing quantum nodes (red) connected by channels (gray). Detectors are for distant and local Bell state measurement. Here we consider a 3-node link containing Alice, Bob and a central FPSA. NVs are Stark-shifted to frequency channels (Ch0-Ch3) at different points in the protocol indicated by step 1-4. Step 1: NV preparation, which contains optical phase stabilization and NV charge/resonance state check. Step 2: Stark shift is applied to set NV optical transition to a spectral channel that communicates with the distant link, and a heralded single photon scheme is applied to generate entanglement. Step 3: Stark shift is applied to set the NV transition to

Ch0 to wait for the other successful distant entanglement on the other side. Step 4: Stark shift is applied to set these two NVs to a shared spectral channel and to make the entanglement between the two local NVs via a Barrett-Kok scheme. **b** $\Gamma_{AB}$ over a L = 1 km channel as a function of the number of qubits for FPSA (red) and MZI tree (blue) architectures. Above 3000 qubits, the entanglement procedure is limited by the channel capacity. **c** $\Gamma_{AB}$ as a function of the number of qubits for a hybridization of MZI and FPSA architecture. **d** Optimal $\Gamma_{AB}$ for different link length and number of qubits. The dotted line shows the largest $\Gamma_{AB}$ for different link length.

photon lifetime. For a 1 km link and 10 ns lifetime, we have $N_{ch} \sim 300$ time-multiplexing channels. Additional frequency channels can further raise the number of time-frequency bins for distant entanglement generation. The Stark shift tuning range, bandwidth of the slow-light effect, qubit linewidth and frequency multiplexer bandwidth limit the number of multiplexing channels. In our device, the regime of the Purcell factor > 10 has a bandwidth of ~200 GHz, setting the total frequency range. Here we assume a 20 GHz bandwidth for the frequency multiplexer based on the potential dense wavelength division multiplexing[54]. Therefore, the time-frequency channel capacity in FPSA is $10 \, t_{link}/t_{ph}$. In the regime that we have fewer qubits than the channel capacity, each qubit pair can effectively generate entanglement independently. Above this number, qubits will compete for channel usage.

The same dynamically tunable operations allow us to immediately attempt local entanglement as soon as distant entanglement is heralded. After a heralding signal from both A and B, we shift both qubits $j, k$ to Ch3 and generate local entanglement as described above. Due to localized independent electric field-based control, we can parallelize local entanglement generation (step 2) while simultaneously attempting entanglement over the long distance (step 1) links using other qubits (e.g. $j, k = j+1, k+1$) rather than requiring sequential operations. The entanglement rate is then mainly limited by the first step.

While increasing qubit number improves $\Gamma_{AB}$ linearly in the ideal case, each additional qubit adds exponential loss to the device as

larger device size leads to transmission $\eta_{wg} \propto \exp(-t_{wg}N)$. The entanglement rate is given by $\Gamma_{AB} \propto N \exp(-t_{wg}N)$, as shown in the blue curve in Fig. 4b for parameters given in Table 1. The FPSA reaches a maximum rate of $\Gamma_{AB} = 2.1 \times 10^4$ ebits/s when the number of qubits is $N = 1824$, after which loss decreases rate exponentially with increasing number of qubits. The red curve shows the rate by a Mach-Zehnder interferometer (MZI) tree architecture as a comparison[55]. Here we assume an 0.4 dB loss per MZI[56,57]. In this regime, the FPSA outperforms the MZI architecture by a factor of ~3. In the limit of large qubit number, the rate scales linearly with $N$. With very large qubit numbers, the MZI architecture could outperform FPSA because it is not suffering from exponential loss. However, the number of parallel qubits is limited by the time-frequency multiplexing channel capacity. For a 1 km link, the channel capacity is $10 \, t_{link}/t_{ph} \sim 3000$ shown in Fig. 4b.

As an extension, we consider a hybridization of FPSA and MZI tree architectures. As shown in Fig. 4c, we divide the qubits into $N_{dev}$ FPSAs connected by an MZI tree. $\Gamma_{AB}$ as a function of the number of qubits with different $N_{dev}$ is shown in Fig. 4c. Taking the optimal $N_{dev}$ for each qubit number, we plot the maximum rate envelope shown in the dashed line. Instead of exponential decay, the optimal envelope asymptotically follows a linear scaling. In this scheme, the rate is limited by time-frequency multiplexing channels capacity shown in the gray region in Fig. 4c. For different link lengths, the channel limit changes, resulting in varied maximum rate as shown in Fig. 4d. The rate

can be straightforwardly increased with additional frequency-multiplexing channels. Alternatively, a fixed number of memories $N$ can be used more efficiently in schemes with a midway entangled photon pair source[58,59], increasing the entanglement rate from $\propto \eta$ to $\sqrt{\eta}$.

## Discussion

We comment briefly on (i) fabrication and (ii) qubit choices of the presented FPSA blueprint. (iii) other design considerations for strain-based FPSA.

(i) Since the diamond can be placed on top of a periodic dielectric perturbation (e.g., by pick-and-place of diamond waveguides[60,61]), enabling the substrate to be designed separately, a number of material choices are available. A potential approach is to produce high-index dielectric fins is to use atomic layer deposition of $HfO_2$/AlN followed by lithography and a lift-off process[62], while some common photonic crystal structure like[63] may be hard to fabricate with diamond due to the difficulty to get a large diamond thin film through undercutting. Moreover, the substrate can be fabricated in CMOS to proximally position the required electrical contacts through a back-end-of-line metallization step. Custom CMOS processes have already been successfully demonstrated for NV spin control via microwave magnetic field interactions[64], whereas the electric field control should be easier as less current is required for a given Rabi frequency. Other materials with high dielectric constant such as barium titanate (BTO, $\varepsilon \sim 7000$) would serve as attractive alternatives to $HfO_2$ as they could also provide electro-optic modulation of traveling modes[65].

(ii) We considered the diamond NV center because of the reported spin-electric field coupling Hamiltonian and high dielectric strength of diamond—but the NV has several drawbacks. Its rather low coupling strength $d'_\perp$ for electric field driving of the $|\pm 1\rangle \leftrightarrow |0\rangle$ spin transition limits the Rabi frequency $f \propto d'_\perp$ that can be achieved without driving up the power dissipation $P_E \propto 1/d'^2_\perp$ or risking electrical breakdown. One promising path to address this challenge is to use global microwave driving on the NV $|\pm 1\rangle \leftrightarrow |0\rangle$ transition so that individual control can be performed on the $|+1\rangle \leftrightarrow |-1\rangle$ transition which has a $\sim 50\times$ stronger coupling to the electric field. Generating large arrays of NV centers with lifetime-limited optical coherence remains an open challenge. NV centers show large spectral diffusion in nanostructures[66,67] but several recent works have improved the performance[68,69]. There are several other ways to deal with the spectral diffusion problem. (1). FPSA can use a feedback system to reduce the NV linewidth[70]. The low capacitance of FPSA allows a short time constant for charging the system, but the feedback will require more measurement time during the quantum entanglement generation process. (2). NVs in larger multi-mode waveguides far from sidewalls may reduce spectral diffusion like previous work[71]. The NV centers can still couple to slow-light mode with large Purcell factor via proper design, although there are multiple modes in the waveguide.

The use of the FPSA architecture can be generalized for other color centers in diamond and emitters in other wide-bandgap materials[6,10–12,30,72], though different properties (e.g., different electric field-spin dipole coupling constants) would require different trade-offs. Emitters in silicon carbide have been demonstrated with a large tuning range by electric field, which can be a promising candidate for FPSA[73].

(iii) For strain-based FPSA, current design launches acoustic waves along the waveguide, inducing a large cross-talk along waveguide direction. Several other designs can be used to localize the acoustic mode like the interdigital transducers (IDT) structures[13].

A co-designed photonic-phononic bandgap structure can also be used to localize acoustic waves[74].

Here we have presented an FPSA architecture that addresses several challenges in the development of scalable quantum networks. We showed that electric field control is beneficial for the individual quantum addressing of dense emitter arrays, as power consumption and crosstalk are significantly reduced in comparison to the magnetic field case. Furthermore, the wide tunability via the Stark effect allows for multi-channel, parallelized optical entanglement schemes that offer improved scaling with number of qubits. Strain field-based FPSA have similar advantages and can be a candidate for color centers with inversion symmetry. Based on these advantages, we expect that FPSA architectures will form the basis of future quantum networking implementations.

## Methods

The electric and strain field calculations are conducted by COMSOL Multiphysics. A Multiphysics containing Electrostatics and Solid Mechanics modules are used to solve the strain field. The optical bandstructure and finite-difference time-domain simulations are performed by MIT Photonic Bands (MPB)[75] and Ansys Lumerical FDTD. The quantum repeater calculation is made by custom MATLAB codes.

## Data availability

All Source data are provided as a Source Data file [https://github.com/hanfengw/FPSA_Source_Data].

## Code availability

The code used to generate data discussed in the manuscript are provided in [https://github.com/hanfengw/FPSA_code].

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

## Acknowledgements

H.W. acknowledges support from the National Science Foundation Center for Ultracold Atoms (NSF CUA). M.E.T. acknowledges support through the Army Research Laboratory ENIAC Distinguished Postdoctoral Fellowship. L.K. acknowledges support through an appointment to the Intelligence Community Postdoctoral Research Fellowship Program at the Massachusetts Institute of Technology, administered by Oak Ridge Institute for Science and Education through an interagency agreement between the U.S. Department of Energy and the Office of the Director of National Intelligence. H.R. acknowledges support from the National Science and Engineering Graduate (NDSEG) fellowship and NSF CUA. D.R.E. acknowledges support from the Bose Research Fellowship, the Army Research Office Multidisciplinary University Research Initiative (ARO MURI) biological transduction program, and the NSF CUA. We thank Ian Christen, Dr. Hyeongrak Choi, Kevin C. Chen and Dr. Lorenzo de Santis for helpful discussions.

## Author contributions

M.E.T. and D.R.E. conceived the project. H.W. and M.E.T made the design and the quantum repeater calculation. H.W. performed the simulations and models. L.K. assisted in optical simulations. H.R. assisted in strain-based FPSA design. H.W. and M.E.T. prepared the manuscript. All authors discussed results and revised the manuscript. M.E.T. and D.R.E. supervised the project.

## Competing interests

The authors declare no competing interests.
