## [Peer Review File · Nature Communications]

REVIEWER COMMENTS

Reviewer #1 (Remarks to the Author):

Wang et al report on an architecture for a quantum network based on nitrogen-vacancy color-centers integrated with dielectric structures designed for combined electrical spin control and optical Purcell enhancement. Overall I find the paper relatively clear yet detailed, and to contain interesting new suggestions towards the larger pursuit of realizing a scalable quantum network. However, while there are certainly novel elements in the work, I find it to be of questionable practical significance due to significant practical open questions with the approach, and do not at this point recommend it for publication in a journal with the impact of Nature Communications.

My main concerns (with suggestions for improvement) include:

1. The NV center has not been shown to work reliably/effectively in nanostructures. I suggest here some possible directions on how the authors can broaden the impact of this work with further investigation:

a. Can this approach be applied to/practical for other color-centers/systems with improved optical coherence in fabricated structures?

b. Maybe you can comment quantitatively on the feasibility for your system to provide high locking bandwidth to stabilize the NV optical transition given the typical noise spectrum in photonic structures?

c. Could the same approach be compatible with NV-cavity efforts involving larger photonic systems that do not require nanofabrication or close proximity to surface, such as fiber Fabry-Perot cavities?

2. Direct electric field driving, to my knowledge, has not been demonstrated with the NV center. It is not clear to me that this proposal is ultimately more practical than an approach to use strain-based driving, which has been experimentally achieved and to the best of my understanding should present similar advantages in terms of cross-talk and heating compared to conventional B-field driving.

In order to recommend this work for publication in Nature Comms I would like to see both points suitably addressed so as to provide a clear roadmap from what has been experimentally demonstrated with NV centers today to the vision presented in this paper. This would include a compelling quantitative argument along any of 1(a-c) such that consistently realizing a 100 MHz NV transition may actually be feasible. I would also need to see experimental evidence of the feasibility of E-field driving (at least in some proof-of-concept format).

In addition to these objections, I have some additional questions that I would like to see answered prior to publication in any form:

3. Can you clarify what the peak electrical current in the system is assuming driving by a 50 ohm source? I am concerned that while the system may reduce the current "on-chip", there would be high currents in other parts of the circuit (control lines, connectors, etc) which are still a part of the cryostat. After all, the wavelength of these systems is on the \sim cm scale which is typically still much smaller than most cryogenic apparatuses. I am worried one may be limited by heat dumped at this components.

4. As I understand it the structure proposed provides delivery of 3 types of fields:

a. Large classical DC fields for stark tuning

b. Large classical MW fields for electron manipulation

c. Quantum optical fields for networking/remote interactions

What is the interaction between these fields? For example, will the large MW fields cause an AC stark shift of the optical transition? this does not seem to be considered. Also, will nonlinearities in the materials cause problems? For example, will the large electric fields from DC/MW tuning cause shifts in the photonic structure?

5. I suggest the authors to stick to a convention of V/m or V/um for clarity.

Reviewer #2 (Remarks to the Author):

In this manuscript, the authors propose 'electric-field programmable spin arrays' (eFPSA) to enable high-speed and relatively large-scale control of NV centers with low cross-talk and power dissipation. They also discuss the performance of coupling the eFPSA to a slow-light wave guide and then utilize it for scalable quantum repeater (QR) architectures. My comments below are mainly about its quantum repeater part (Sec. IIC).

As far as I understand, typically the diamond NV center contains only one electron spin, acting as an optical interface, and one or several nuclear spins, acting as memories. This specific feature, only one electron spin, limits the operation efficiency and makes it difficult to be scalable since most operations have to be performed through electron spins, which involves state initializations and mappings back and forth. In this manuscript, by fabricating lots of spins into a single diamond waveguide (though I am wondering if this is feasible), the authors try to use the concept of multiplexing to make QR more efficient. However, some points are described vaguely and thus hinder understanding.

1. In step (1) of Sec. IIC, after the establishment of entanglement between A(B) and the corresponding electron spins, the state is mapped to nuclear spins. However, in step (2), local entanglement swapping (ES) is performed on electron spins immediately without mentioning the mapping. This should be clarified.
2. It seems that, though electrons are integrated inside a single diamond now, ES is still done with the help of photons, which is a probabilistic process. It's helpful if the authors can elaborate on how Barrett-Kok scheme is implemented in this case. And how this process can be parallelized to make it almost deterministic is also confusing.
3. In the caption of Fig4(a), the specific meaning of step 1-4 confuses me. It would be better to give more details.
4. The background introduction in Sec. I is very limited. For readers who are not an expert in the exact field, it is not enough to tell why "the spin-electric field coupling of the CC ground state" can be "a fundamentally different approach" (line 26), and thus hinder the conclusion if the paper provides the sort of significant advance in the particular area.

Reviewer #3 (Remarks to the Author):

The authors proposed a quantum repeater architecture based on densely-packed diamond NV centers in a programmable electrode array with a photonic waveguide structure to achieve all-to-all connectivity to increase the capacity of quantum repeaters required for scalable networking. The electric-field programmable spin array enables high-speed spin control of individual NVs with low cross-talk and power dissipation. They evaluated the performance in comparison to a routing tree design and showed an increased entanglement generation rate.

I was very much impressed by the great performance of the proposed scheme to extremely enhance the repeater capacity. The authors described not only the proposed architecture but also a novel device structure in detail. Although the components of the proposed device are not necessarily original, their device simulations are well convincing to show the all-to-all connectivity desired for constructing high-performance quantum repeaters.

Although the proposed device is nicely designed, I have some concerns about the feasibility of the nanostructure as shown below.

The authors should comment on the efficiency of generating entanglement between NV centers placed in the photonic waveguide. I would imagine that the efficiency might drop significantly before the number of qubits reaches the maximum of 3000 qubits where the channel capacity limits. I would like to know why the authors chose this type of photonic bandgap structure instead of the typical photonic crystal structure.

It seems that NV centers must be precisely positioned as designed to reduce the cross-talk as the authors claim. The authors should clarify the tolerance of the NV position to achieve the cross-talk

fidelity $f_{CE} > 0.999$ to show the feasibility of the proposal. Dimensions of the device structure such as Fig. 1a, Fig. 2ab, and Fig. 3ac, look distorted too much, misleading to the feasibility of the device fabrication. From Table 1, the aspect ratio of the diamond waveguide should be around 1:4, longer in height. The authors should clarify the structure parameter in the figure. Otherwise, some parameters such as electrode spacing "a" could be misunderstood.

I noticed that this manuscript lacks some references as shown below.

Since Electrically Driven Spin Resonance (EDSR) has been a hot topic for over ten years, the authors should refer to some pioneering papers such as the followings. Among them, especially Klimov in David Awschalom's group experimentally demonstrated EDSR by using color centers in SiC with the same method as this manuscript and Klimov mentioned that their method applies to NV centers in the paper.

M. Pioro-Ladrière et. al., "Electrically driven single-electron spin resonance in a slanting Zeeman field", *Nature Physics*, 4, 776 (2008)

P. V. Klimov et. al. "Electrically Driven Spin Resonance in Silicon Carbide Color Centers", *Phys. Rev. Lett.* 112, 087601 (2014).

A. Corna et. al. "Electrically driven electron spin resonance mediated by spin-valley-orbit coupling in a silicon quantum dot", *npj Quantum Information*, 4, 6 (2018).

The authors referred Barrett-Kok scheme for entanglement generation between wavelength-tuned NVs in a cavity. I assume that the scheme requires homodyne measurement with a beam splitter and two single-photon detectors. I would like to know a more realistic device image for the homodyne measurement.

The authors should also refer to the following paper by Gerhard Rempe's group, which experimentally demonstrated entanglement generation between two atoms in a cavity using a different scheme without requiring homodyne measurement.

S. Welte et. al., "Cavity Carving of Atomic Bell States", *Phys. Rev. Lett.* 118, 21503 (2017).

The authors should also refer to the following papers since they are closely related to the proposed protocol.

Y. Sekiguchi et. al., "Optically addressable universal holonomic quantum gates on diamond spins", DOI:10.21203/rs.3.rs-1461563/v1.

Y. Sekiguchi et. al., "Backward propagating quantum repeater protocol with multiple quantum memories", arXiv:2205.04243, DOI:10.21203/rs.3.rs-1723443/v1.

I noticed some errors or typos as shown below.

Please check Eq. (1). It seems that the authors copied from Ref. 14, but the prime seems to be wrongly positioned.

Please also check Eq. (2). The position of the prime might be opposite corresponding to Eq. (1).

Electric field-driven Rabi frequencies 0.13 GHz for $\{+1, -1\}$ and 1.9 MHz for $\{+-1, 0\}$ should also be checked relating to the above.

Please clarify which parameter the authors used in the claim "A resonant electric field of 10^7 V/m is needed for reaching this gate fidelity (over 0.99)". It is not clear to me which basis $\{+1, -1\}$ or $\{+-1, 0\}$ the authors assumed for the gate operation.

In Fig. 1a, the position of "SiO₂" should be moved since it looks pointing at the NV center.

The second transition dipole μ_2 [-1, -2, 1] seem wrong since it is not perpendicular to the NV axis z [1,1,1] and first transition dipole μ_1 [1, -1, 0].

Fig. 3b is hard to understand. I do not understand what dotted lines and "bandgap" exactly mean. The authors should clarify the position of the photonic bandgap in the photonic band structure in Fig. 3a.

Before recommending this manuscript for publication in Nature Communications, I would like to have the author's responses about the above concerns and possible corrections.

RESPONSE TO REVIEWERS' COMMENTS

Reviewer #1 (Remarks to the Author):

Wang et al report on an architecture for a quantum network based on nitrogen-vacancy color-centers integrated with dielectric structures designed for combined electrical spin control and optical Purcell enhancement. Overall I find the paper relatively clear yet detailed, and to contain interesting new suggestions towards the larger pursuit of realizing a scalable quantum network. However, while there are certainly novel elements in the work, I find it to be of questionable practical significance due to significant practical open questions with the approach, and do not at this point recommend it for publication in a journal with the impact of Nature Communications.

We appreciate that the Reviewer found the paper clear, detailed, interesting, and novel. We specifically address the Reviewer's concerns about the practical significance below.

My main concerns (with suggestions for improvement) include:

1. The NV center has not been shown to work reliably/effectively in nanostructures. I suggest here some possible directions on how the authors can broaden the impact of this work with further investigation:

The Reviewer is correct to note that the NV does have issues with operation in nanostructures - in particular spectral instability (diffusion) that limits photon indistinguishability. We discuss this in the manuscript where we state: "NV centers also show large spectral diffusion in nanostructures (Rodgers et al. 2021; Faraon et al. 2012) but several recent works have improved the performance (Orphal-Kobin et al. 2022; Chakravarthi et al. 2021)."

We used the NV as an example system in large part due to practicality considerations. Quantum network protocols and associated operations (nuclear spin gates for example) have been demonstrated for the NV where they have not for other quantum emitters. Choosing another system without the extensive body of work and literature that the NV has could open up new practicality concerns rather than alleviate them. For the purposes of illustrating the FPSA concepts, we therefore used the NV as a clean example without requiring extrapolation into the unknown.

We also note that several groups are improving upon the NV in nanostructured environments (see (Orphal-Kobin et al. 2022; Chakravarthi et al. 2021)), and so there is hope that the NV will overcome the issues it has in the present day.

a. Can this approach be applied to/practical for other color-centers/systems with improved optical coherence in fabricated structures?

We thank the Reviewer for pointing this out. Applying this work to other quantum emitters is a natural extension, which we addressed briefly in the discussion:

'Beyond NV centers in diamond, the FPSA architecture applies to other color centers in diamond and other wide-bandgap materials (Koehl et al. 2011; Sipahigil et al. 2016; De

Santis et al. 2021; Piore-Ladrière et al. 2008; Klimov et al. 2014; Corna et al. 2018), though different properties (e.g., different electric field-spin dipole coupling constants) would require different trade-offs. Group-IV centers in diamond have drawn interest due to their inversion symmetry and optical properties (Bradac et al. 2019). Emitters in silicon carbide have been demonstrated with a large tuning range by an electric field, which can be a promising candidate for FPSA (Falk et al. 2014). ’

Following the Reviewer's suggestion, with a particular mind to inversion-symmetric group IV centers in diamond, we have expanded this discussion significantly and added additional device designs that extended the spin array concept to piezomechanical driving. The origin of the improved optical coherence in these quantum emitters is insensitivity to electric fields, which requires a different addressing modality (strain), but one that indeed fits nicely in the FPSA framework. We have added a section entitled “Strain-driven quantum gates” that shows piezoelectric control of the optical transition frequency and spin degree of freedom, while integrated with a slow-light waveguide: in short, a strain-based FPSA. We hope this section will show the potential of the FPSA for this class of emitters.

Change to text: We add lines 63-77, 150-179, 191-198, and 225-232 in the draft. Fig. 1 and Fig. 2 are also changed. Appendix VII. Strain tuning design and Appendix VIII. Spectral addressing by strain field are also added for more details.

b. Maybe you can comment quantitatively on the feasibility for your system to provide high locking bandwidth to stabilize the NV optical transition given the typical noise spectrum in photonic structures?

We thank the Reviewer for raising the point of feedback bandwidth. In previous work (Acosta et al. 2012), the ZPL emission frequency can be controlled over a range of 300 GHz and the single scan linewidth can be reduced to 16 MHz using the feedback system. In FPSA, because we have a very low capacitor $C \sim 0.05$ fF, the time constant for the circuit is around $1e-15$ s, so the charging speed will be very high and finally should be limited by the measurement time during the quantum entanglement generation process. We add texts in the discussion part.

Change to text: There are several other ways to deal with the spectral diffusion problem. (1). FPSA can use a feedback system to reduce the NV linewidth (Acosta et al. 2012). The low capacitance of FPSA allows a short time constant for charging the system, but the feedback will require more measurement time during the quantum entanglement generation process.

c. Could the same approach be compatible with NV-cavity efforts involving larger photonic systems that do not require nanofabrication or close proximity to surface, such as fiber Fabry-Perot cavities?

We thank the Reviewer for raising the point of larger photonic systems. The FPSA concept of individual control and frequency multiplexing can be extended to other

systems, but there are several potential issues. For a fiber Fabry-Perot system such as (Ruf et al. 2021), the quantum emitters only exist in one spatial location along the cavity mode. As the FPSA relies on spatially-distinct control, that aspect would require extensive redesign. Second, cavity systems are spectrally narrow. The frequency multiplexing will likely be limited. Following the slow-light waveguide concept, however, larger waveguides that keep the NVs further from sidewalls might be possible - certainly in a multi-mode waveguide.

Change to text: (2). NVs in larger multi-mode waveguides far from sidewalls may reduce spectral diffusion like previous work (Ruf et al. 2021). The NV centers can still couple to slow-light mode with high group velocity via proper design although there are multi-modes in the waveguide.

2. Direct electric field driving, to my knowledge, has not been demonstrated with the NV center. It is not clear to me that this proposal is ultimately more practical than an approach to use strain-based driving, which has been experimentally achieved and to the best of my understanding should present similar advantages in terms of cross-talk and heating compared to conventional B-field driving.

We thank the Reviewer for bringing up the two points of the NV electrically-driven spin resonance and the potential for strain-based driving. As mentioned above, we have used this opportunity to add a new section describing how a piezoelectric strain drive can be implemented in an analogous device.

As to electric driving, we have presented the electric field Hamiltonian of the NV which shows the prospects for driving. As magnetic driving is stronger for most structures (those that do not concentrate electric field as the FPSA does), differentiating experimentally between magnetic and electric driving is difficult. Based on our simulation, the magnetic field is low for FPSA and cannot drive a large Rabi frequency. We note that electric driving has been explored in SiC with a very similar Hamiltonian (also C_{3v} symmetry): (Klimov et al. 2014)

$$H = (hD + d_{\parallel}E_z)\sigma_z^2 + g\mu_B\boldsymbol{\sigma}\cdot\mathbf{B} - d_{\perp}E_x(\sigma_x^2 - \sigma_y^2) + d_{\perp}E_y(\sigma_x\sigma_y + \sigma_y\sigma_x),$$

which indicates that this approach is possible for NV centers.

Based on the simulation results, E-field and strain-field-based FPSA have different merits. Although a smaller voltage is used for strain-field-based FPSA, the localization of the strain field is worse than the electric field (Fig. 2) as an acoustic wave propagates along the diamond waveguide. For DC strain tuning, the strain field has 5 components, which makes it hard to eliminate cross-talk and limits the DC shift range. In Appendix VII,

an optimization method is used to sacrifice parts of the cross-talk fidelity to get a relatively large DC strain tuning.

Change to text: As noted previously, we have added a new geometry and discussion of strain-mediated programmable spin arrays.

In order to recommend this work for publication in Nature Comms I would like to see both points suitably addressed so as to provide a clear roadmap from what has been experimentally demonstrated with NV centers today to the vision presented in this paper. This would include a compelling quantitative argument along any of 1(a-c) such that consistently realizing a 100 MHz NV transition may actually be feasible. I would also need to see experimental evidence of the feasibility of E-field driving (at least in some proof-of-concept format).

We thank the Reviewer again for their thoughtful comments and the opportunity to improve the manuscript. We hope that the responses noted above will convince the Reviewer that the FPSA architecture is a pathway toward scalable quantum repeaters.

In addition to these objections, I have some additional questions that I would like to see answered prior to publication in any form:

3. Can you clarify what the peak electrical current in the system is assuming driving by a 50 ohm source? I am concerned that while the system may reduce the current “on-chip”, there would be high currents in other parts of the circuit (control lines, connectors, etc) which are still a part of the cryostat. After all, the wavelength of these systems is on the ~ cm scale which is typically still much smaller than most cryogenic apparatuses. I am worried one may be limited by heat dumped at this components.

We thank the Reviewer for bringing this up, which may not have been clear in the text. The peak electrical current in the system for an $F = 0.99$ for the double quantum gate is around 98 mA with a 50-ohm source. And for the strain-based FPSA, the maximum current for $F = 0.99$ is 1.4 mA.

Change to text: We added the maximum current in the system at lines 253-254 and lines 268-281

4. As I understand it the structure proposed provides delivery of 3 types of fields:

- a. Large classical DC fields for stark tuning
- b. Large classical MW fields for electron manipulation
- c. Quantum optical fields for networking/remote interactions

What is the interaction between these fields? For example, will the large MW fields cause an AC stark shift of the optical transition? this does not seem to be considered. Also, will nonlinearities in the materials cause problems? For example, will the large electric fields from DC/MW tuning cause shifts in the photonic structure?

We thank the Reviewer for this opportunity to clarify this segment.

There are many reference papers about the combination of optical fields and MW fields or AC Stark shifts caused by MW fields for NV centers (Sekiguchi et al. 2022). However, in our case, we will not apply the AC and DC fields at the same time. For example, we will first use an AC electric field to prepare the state and then turn off the AC electric field and turn on the DC electric field for Stark tuning.

For the nonlinearities in the materials, HfO₂ has $\chi(2)$, and the electrical-optic (EO) coefficient is ~ 5.7 pm/V (Kondo et al. 2021). This means a 10 MV/m electric field will give us a local refractive index change of 1.9 ± 0.0002 . This local refractive index is not large enough to cause shifts in the photonic structure.

The EO coefficient of AlN is ~ 0.67 pm/V (Gräupner et al. 1992) so the refractive index change is smaller than HfO₂.

5. I suggest the authors stick to a convention of V/m or V/μm for clarity.

We thank the Reviewer for pointing this out, and have chosen V/μm throughout as it matches the actual scale of the device.

Change to text: We change all units to V/μm.

Reviewer 2

In this manuscript, the authors propose ‘electric-field programmable spin arrays’ (eFPSA) to enable high-speed and relatively large-scale control of NV centers with low cross-talk and power dissipation. They also discuss the performance of coupling the eFPSA to a slow-light wave guide and then utilize it for scalable quantum repeater (QR) architectures. My comments below are mainly about its quantum repeater part (Sec. IIC).

As far as I understand, typically the diamond NV center contains only one electron spin, acting as an optical interface, and one or several nuclear spins, acting as memories. This specific feature, only one electron spin, limits the operation efficiency and makes it difficult to be scalable since most operations have to be performed through electron spins, which involves state initializations and mappings back and forth. In this manuscript, by fabricating lots of spins into a single diamond waveguide (though I am wondering if this is feasible), the authors try to use the concept of multiplexing to make QR more efficient. However, some points are described vaguely and thus hinder understanding.

1. In step (1) of Sec. IIC, after the establishment of entanglement between A(B) and the corresponding electron spins, the state is mapped to nuclear spins. However, in step (2), local entanglement swapping (ES) is performed on electron spins immediately without mentioning the mapping. This should be clarified.

We thank the Reviewer for this opportunity to clarify this segment. In step (2), after the local entanglement between two electron spins in FPSA is established. A CNOT gate is performed between electron spin and nuclear spin to establish the local entanglement, then Bell measurements in the electron-nuclear spin basis of the memories j and k swap the local entanglement to distant entanglement of A and B after subsequent feed-forward (Degen et al. 2021; Pfaff et al. 2012).

Change to text: A CNOT gate is performed between electron spin and nuclear spin to establish the local entanglement, then Bell measurements in the electron-nuclear spin basis of the memories j and k swap the local entanglement to distant entanglement of A and B after subsequent feed-forward (Degen et al. 2021; Pfaff et al. 2012).

2. It seems that, though electrons are integrated inside a single diamond now, ES is still done with the help of photons, which is a probabilistic process. It's helpful if the authors can elaborate on how Barrett-Kok scheme is implemented in this case. And how this process can be parallelized to make it almost deterministic is also confusing.

We thank the Reviewer for this opportunity to clarify this segment. We agree with the reviewer that this process is still done with the help of photons, as shown in previous papers:

Bernien, Hannes, et al. "Heralded entanglement between solid-state qubits separated by three metres." *Nature* 497.7447 (2013): 86-90.

Humphreys, Peter C., et al. "Deterministic delivery of remote entanglement on a quantum network." *Nature* 558.7709 (2018): 268-273.

In those papers, they treated this probabilistic process as deterministic by considering a fixed number of trials, resulting in a mixed-state output. In our case, it is a probabilistic process where we calculated the average number of heralded entanglement generation events per unit of time, with fidelity $F = 0.99$.

We implement a Barrett-Kok scheme that works like the reference paper (Bernien et al. 2013) in order to generate entanglement between two emitters in the FPSA. We prepare the states to be $1/\sqrt{2} (|0\rangle + |1\rangle)$ and excite $|0\rangle$ to $|e\rangle$ for both emitters, by a resonant laser that is orthogonal to the slow light mode (e.g. from the top). The emitters then relax, and if the photons emitted are indistinguishable, detection of a heralding photon will project the state to $1/\sqrt{6} (|01\rangle + |10\rangle + 2|11\rangle)$. For the consideration of eliminating the probability of two-photon emission, we flip the spins and try to get a heralding photon again (projecting into the single-excitation subspace) - if this is successful, we consider this to be a heralded entanglement generation event. If not, we repeat the process.

This process can be parallelized because each qubit can be locally-controlled by electric fields or strain fields, and can be moved to different frequency or time multiplexing

channels, enabling preparing the states separately for all emitters. The number of successful entanglement generation events in any given iteration is still random, but we report the average number of successes per time. We have clarified the text on this point.

Change to text: Added “average“ to entanglement rate in line 412, and additional simulation details in the Supplementary Information.

3. In the caption of Fig4(a), the specific meaning of step 1-4 confuses me. It would be better to give more details.

We thank the Reviewer for pointing this out. Steps 1-4 shown in Fig. 4(a) are in the time domain. Step 1: NV preparation, which contains optical phase stabilization and NV charge/resonance state check. Step 2: Stark shift is applied to set NV optical transition to a spectral channel that communicates with the distant link, and a heralded single photon scheme is applied to generate entanglement. Step 3: Stark shift is applied to set the NV transition to the bandgap to wait for the other successful distant entanglement on the other side. Step 4: Stark shift is applied to set these two NVs to a shared spectral channel to make the entanglement between the two local NVs via a Barrett-Kok scheme.

Change to text: We add an additional caption for Fig. 4(a):

Step 1: NV preparation, which contains optical phase stabilization and NV charge/resonance state check. Step 2: Stark shift is applied to set NV optical transition to a spectral channel that communicates with the distant link, and a heralded single photon scheme is applied to generate entanglement. Step 3: Stark shift is applied to set the NV transition to the bandgap to wait for the other successful distant entanglement on the other side. Step 4: Stark shift is applied to set these two NVs to a shared spectral channel to make the entanglement between the two local NVs via a Barrett-Kok scheme.

4. The background introduction in Sec. I is very limited. For readers who are not an expert in the exact field, it is not enough to tell why “the spin-electric field coupling of the CC ground state” can be “a fundamentally different approach” (line 26), and thus hinder the conclusion if the paper provides the sort of significant advance in the particular area.

We thank the Reviewer for the comments. Here we rewrite part of the introduction to make this clear.

Change to text: Previous work achieved localized control using a magnetic field with a spatial gradient (Bodenstedt et al. 2018), or by producing a spatially-varying detuning of the CC resonant frequency using a gradient magnetic (Jakobi et al. 2017) or optical (Sekiguchi et al. 2022) field combined with global magnetic addressing. Here, we propose a fundamentally different approach that uses highly localized fields - either strain or electric, depending on the CC of choice - which can be driven capacitively for low power dissipation. Electric field-based spin control has been proposed in several systems (Pioro-Ladrière et al. 2008; Klimov

et al. 2014; Corna et al. 2018), while strain driving has been demonstrated for many CC systems (Maity et al. 2020; Udvarhelyi et al. 2018)

Reviewer 3

The authors proposed a quantum repeater architecture based on densely-packed diamond NV centers in a programmable electrode array with a photonic waveguide structure to achieve all-to-all connectivity to increase the capacity of quantum repeaters required for scalable networking. The electric-field programmable spin array enables high-speed spin control of individual NVs with low cross-talk and power dissipation. They evaluated the performance in comparison to a routing tree design and showed an increased entanglement generation rate.

I was very much impressed by the great performance of the proposed scheme to extremely enhance the repeater capacity. The authors described not only the proposed architecture but also a novel device structure in detail. Although the components of the proposed device are not necessarily original, their device simulations are well convincing to show the all-to-all connectivity desired for constructing high-performance quantum repeaters.

We would like to thank the Reviewer for the thoughtful and thorough review of the manuscript, and we are grateful for the positive assessment and interest in the work. We address the comments point by point below.

The authors should comment on the efficiency of generating entanglement between NV centers placed in the photonic waveguide. I would imagine that the efficiency might drop significantly before the number of qubits reaches the maximum of 3000 qubits where the channel capacity limits.

We thank the Reviewer for this opportunity to clarify this segment. We consider two ways that decrease entanglement generation efficiency. Adding more periods will introduce more photon loss. We model it as a photon loss/period, which is $8e-4$ dB/period (or 44 dB/cm) based on FDTD simulation.

The current manuscript uses total rate as a figure of merit rather than efficiency, which could be interpreted as a rate per qubit. To clarify this point, we added another figure in the appendix that plots the rate per qubit, and the diminishing returns of device size are more clear. This figure shows when the number of qubits is larger than 1764, the entanglement rate begins to decrease when we add more qubits because of the added loss.

For the hybridization of FPSA and MZI tree architectures, we have a paragraph to talk about the scaling consideration: “As an extension, we consider a hybridization of FPSA and MZI tree architectures. As shown in Fig. 4(c), we divide the qubits into N_{dev} FPSAs connected by an MZI tree. Γ_{AB} as a function of the number of qubits with different N_{dev} is shown in Fig. 4(c). Taking the optimal N_{dev} for each qubit number, we plot the maximum rate envelope shown in the dashed line. Instead of exponential decay, the optimal envelope asymptotically follows a linear scaling.”

Fig. S5 also shows the decreased efficiency when adding more qubits for the hybrid structure.

Change to text: We have added Appendix Fig. S5 to clarify this segment, with the caption: Increase entanglement rate per added qubit for FPSA (blue), MZI tree (red) and hybrid structure (yellow).

I would like to know why the authors chose this type of photonic bandgap structure instead of the typical photonic crystal structure.

We thank the Reviewer for this opportunity to clarify this point. For the photonic crystal structure design, we considered many photonic crystal structures but we found that this one is more suitable for the FPSA because of the merits listed below.

(1). There are many types of photonic crystal cavity structures proposed in previous works such as:

Evans, Ruffin E., et al. "Photon-mediated interactions between quantum emitters in a diamond nanocavity." *Science* 362.6415 (2018): 662-665.

However, in our proposal, we want many qubits coupled in the same waveguide mode and they can interact via a single waveguide mode. So a slow light waveguide is more suitable in our case.

(2). The fin structure with a large dielectric constant can localize the electric field. In the electric field-based FPSA design. The HfO₂ fin structure not only acts as the photonic crystal structure but also acts as an electric field concentrator. These HfO₂ E-field concentrators with a high DC refractive index of ~ 23 will decrease the cross-talk and increase the electric field in diamonds. In Appendix II, we compare the cases with/without fin structures. Based on COMSOL simulations, The field maximum and the spatial refinement are reduced by a factor of 1.7, increasing cross-talk fidelity from 0.66 to 0.92.

(3). In our generalized device with piezoelectric structures, we use piezoelectric material (e.g. AlN) instead of HfO₂. Based on our calculation, this structure can transfer the strain to the diamond and get an efficient strain tuning on defects in the diamond.

(4). This structure can be easily fabricated by etching AlN/HfO₂ and transferring the diamond waveguide using the pick-and-place process. Some other photonic crystal waveguide structures such as:

Scarpelli, L., et al. "99% beta factor and directional coupling of quantum dots to fast light in photonic crystal waveguides determined by spectral imaging." *Physical Review B* 100.3 (2019): 035311.

are hard to fabricate with diamonds because it is hard to get a large diamond thin film through undercutting.

Change to text: We added Supplementary I: Device design considerations to discuss all things listed above. In the discussion part, we add the fabrication consideration. In lines 55-58, we add the advantages of the slow-light waveguide.

It seems that NV centers must be precisely positioned as designed to reduce the cross-talk as the authors claim. The authors should clarify the tolerance of the NV position to achieve the cross-talk fidelity $f_{CE} > 0.999$ to show the feasibility of the proposal.

We thank the Reviewer for pointing that out. To make the statement more clear, we change it to $F_{CE} > 0.99$ so that it can be compared directly with dephasing fidelity $F_{dephasing} = 0.99$. The tolerance of the NV position to achieve this fidelity is 18 nm (73 nm) for the first (second) nearest neighbors shown in Fig. 2(c). For the newly-added strain driving geometry, the NV position's tolerance along the z-axis to achieve the cross-talk fidelity $F_{CE} > 0.99$ is $\Delta z = 110$ nm. We add a cut line to show the F_{CE} line in Fig. 2(c) and Fig. 2(f) for further clarification.

Additionally, the CE still works if NVs are not in that position. Our cross-talk elimination process still works if NVs are not positioned exactly with the photonic crystal structures. In the real device, we can apply voltage on each electrode to measure Rabi frequencies and calculate the map matrix by experiment results.

Change to text: We added a cut line to show the F_{CE} line in Fig. 2(c) and Fig. 2(f). Discussions about position tolerance are added.

Dimensions of the device structure such as Fig. 1a, Fig. 2ab, and Fig. 3ac, look distorted too much, misleading to the feasibility of the device fabrication. From Table 1, the aspect ratio of the diamond waveguide should be around 1:4, longer in height. The authors should clarify the structure parameter in the figure. Otherwise, some parameters such as electrode spacing "a" could be misunderstood.

We thank the Reviewer for pointing this out. There is a typo on the table, the diamond waveguide width is $0.364 \mu\text{m}$ and the thickness is $0.091 \mu\text{m}$. The aspect ratio of the diamond waveguide should be around 1:4, longer in width, which fits well with Fig.

1a, Fig. 2ab and Fig. 3ac. This diamond can be well seated on an HfO₂ array in fabrication.

Change to text: We change Table I for the right dimensions.

I noticed that this manuscript lacks some references as shown below.

Since Electrically Driven Spin Resonance (EDSR) has been a hot topic for over ten years, the authors should refer to some pioneering papers such as the followings. Among them, especially Klimov in David Awschalom's group experimentally demonstrated EDSR by using color centers in SiC with the same method as this manuscript and Klimov mentioned that their method applies to NV centers in the paper.

M. Pioro-Ladrière et. al., "Electrically driven single-electron spin resonance in a slanting Zeeman field", *Nature Physics*, 4, 776 (2008)

P. V. Klimov et. al. "Electrically Driven Spin Resonance in Silicon Carbide Color Centers", *Phys. Rev. Lett.* 112, 087601 (2014).

A. Corna et. al. "Electrically driven electron spin resonance mediated by spin–valley–orbit coupling in a silicon quantum dot", *npj Quantum Information*, 4, 6 (2018).

We thank the Reviewer for providing those papers; they are relevant to our work. Especially P. V. Klimov et. al. "Electrically Driven Spin Resonance in Silicon Carbide Color Centers", *Phys. Rev. Lett.* 112, 087601 (2014). We have added it to the reference list.

Change to text: Electric field-based spin control has been proposed in several systems (Pioro-Ladrière et al. 2008; Klimov et al. 2014; Corna et al. 2018)

The authors referred Barrett-Kok scheme for entanglement generation between wavelength-tuned NVs in a cavity. I assume that the scheme requires homodyne measurement with a beam splitter and two single-photon detectors. I would like to know a more realistic device image for the homodyne measurement.

We thank the Reviewer for this opportunity to clarify this segment. We agree that Barrett-Kok should have a beam splitter and two single-photon detectors. To clarify this, we added the detectors and beamsplitters for the distant link in Fig. 4(a). For local entanglement generation, we can add two detectors and a beamsplitter on both sides of the waveguide in the normal case. However, the waveguide itself here can act as a beamsplitter which eliminates the which-path information. We can just add two photodetectors on both sides. We add the photodetectors in Fig. 4(a) to make it to be more clear.

Based on the added detectors and beamsplitters, we added more considerations and made our calculation more accurate.

1. For step 1, we assume a single-side detection for step 1 (Entanglement generation between distant links A and B and quantum repeater) but assume a double-side detection for step 2 (Local entanglement generation). As shown in Fig. 4(a) below, for qubit j (k) on Ch1 (Ch2) we only consider the left (right) emission, which means we only make use of half of the signal. For the local entanglement generation process at Ch3, we assume a detector on both sides.
2. We consider the mismatch of the number of entangled qubits on the left and right sides. The number of entangled pairs for Step 1 is $\min(N_{\text{left}}, N_{\text{right}})$. This number mismatch will change the entanglement rate for the low-number qubits regime.

After adding those considerations, the maximum rate does not change but the number of qubits for the maximum changes. We added Supplementary VI Quantum repeater simulation to include all the simulation details.

Change to text: We redrew Fig. 4(a) and added the detectors to clarify. Fig. 4(b) - Fig. 4(d) are slightly changed. We also add a supplementary paragraph about the quantum repeater simulation details.

The authors should also refer to the following paper by Gerhard Rempe's group, which experimentally demonstrated entanglement generation between two atoms in a cavity using a different scheme without requiring homodyne measurement.

S. Welte et. al., "Cavity Carving of Atomic Bell States", Phys. Rev. Lett. 118, 21503 (2017).

We thank the Reviewer for providing this paper. Many schemes have been proposed for entanglement generation (Welte et al. 2017; Kimiaee Asadi, Wein, and Simon 2020; Barrett and Kok 2005). Here we choose this scheme because it works for waveguide structure. Rather than coupling all emitters in a cavity mode, we need to couple emitters in different modes to realize multiplexing.

Change to text: Step 1: Distant entanglement between A(B) and electron spin $|j_e\rangle(|k_e\rangle)$ in the FPSA. Many schemes are proposed for entanglement generation within cavities and waveguides (Welte et al. 2017; Kimiaee Asadi et al. 2020; Barrett and Kok 2005).

The authors should also refer to the following papers since they are closely related to the proposed protocol.

Y. Sekiguchi et. al., "Optically addressable universal holonomic quantum gates on diamond spins", DOI:10.21203/rs.3.rs-1461563/v1.

Y. Sekiguchi et. al., "Backward propagating quantum repeater protocol with multiple quantum memories", arXiv:2205.04243, DOI:10.21203/rs.3.rs-1723443/v1.

We thank the Reviewer for providing these two papers. These papers are quite relevant to our proposed protocol. The first paper proposed high-fidelity operations for single qubits through amplitude-alternating pulses, enabling individual spin control. The second paper proposed a quantum repeater protocol based on backward propagating photon emission and absorption without requiring full connectivity. We have added both papers to our reference list.

Change to text: Previous work achieved localized control using a magnetic field with a spatial gradient (Bodenstedt et al. 2018), or by producing a spatially-varying detuning of the CC resonant frequency using a gradient magnetic (Jakobi et al. 2017) or optical (Sekiguchi et al. 2022) field combined with global magnetic addressing. Here, we propose a fundamentally different approach that uses highly localized fields - either strain or electric, depending on the CC of choice - which can be driven capacitively for low power dissipation. Electric field-based spin control has been proposed in several systems (Pioro-Ladrière et al. 2008; Klimov et al. 2014; Corna et al. 2018), while strain driving has been demonstrated for many CC systems (Maity et al. 2020; Udvarhelyi et al. 2018)

Change to text: Many quantum repeater protocols are proposed in previous works (Sekiguchi, Okumura, and Kosaka 2022; Rozpędek et al. 2019; Lee et al. 2020)

Many quantum repeater protocols have been proposed in previous works, including the use of quantum emitters as quantum memories (Rozpędek et al. 2019), and multiplexing schemes that achieve all-to-all connectivity (Sekiguchi, Okumura, Lee et al. 2020).

I noticed some errors or typos as shown below.

We thank the Reviewer for pointing out those errors and typos. We modify them and make more clarification about the details.

Please check Eq. (1). It seems that the authors copied from Ref. 14, but the prime seems to be wrongly positioned.

Please also check Eq. (2). The position of the prime might be opposite corresponding to Eq. (1).

Electric field-driven Rabi frequencies 0.13 GHz for $\{+1, -1\}$ and 1.9 MHz for $\{+-1, 0\}$ should also be checked relating to the above.

We thank the Reviewer for pointing this out. We make a double check, this is a typo in Eq. (1). The prime is wrongly positioned. The position of the prime in Eq. (2) is not opposite.

Change to text: We modify Eq. (1) to add the prime in the right position.

Please clarify which parameter the authors used in the claim “A resonant electric field of 10^7 V/m is needed for reaching this gate fidelity (over 0.99)”. It is not clear to me which basis $\{+1, -1\}$ or $\{+-1, 0\}$ the authors assumed for the gate operation.

Here we are referring to $\{+1, -1\}$, we change the sentence to make sure it is more clear.

Change to text: For double quantum transition, a resonant electric field of 10 V/mum is needed for reaching this gate fidelity.

In Fig. 1a, the position of “SiO₂” should be moved since it looks pointing at the NV center.

Change to text: We redrew Fig. 1 to make sure it is clear for reading

The second transition dipole μ_2 [-1, -2, 1] seem wrong since it is not perpendicular to the NV axis z [1,1,1] and first transition dipole μ_1 [1, -1, 0].

This is a typo here, we change it to be [-1, -1, 2].

Change to text: $\vec{\mu}_2 = [\bar{1}\bar{1}2]$

Fig. 3b is hard to understand. I do not understand what dotted lines and “bandgap” exactly mean. The authors should clarify the position of the photonic bandgap in the photonic band structure in Fig. 3a.

We thank the referee for pointing this out. We found that the “bandgap” argument may be confusing and limited. In Fig. 3b, we do not need the NV positioned in bandgap for Ch0, the Ch0 can be in any arbitrary frequency as long as the Purcell factor of this frequency is low. Therefore, we generalize our conclusion and get rid of all the bandgap arguments.

Change to text: We change Fig. 3a and Fig. 3b to make them clear.

Before recommending this manuscript for publication in Nature Communications, I would like to have the author's responses about the above concerns and possible corrections.

We thank the Reviewer again for their thoughtful comments and positive assessment and interest in the work. We hope that the responses noted above will solve the concerns.

References:

- Acosta, V. M., C. Santori, A. Faraon, Z. Huang, K-M C. Fu, A. Stacey, D. A. Simpson, et al. 2012. "Dynamic Stabilization of the Optical Resonances of Single Nitrogen-Vacancy Centers in Diamond." *Physical Review Letters* 108 (20): 206401.
- Barrett, Sean D., and Pieter Kok. 2005. "Efficient High-Fidelity Quantum Computation Using Matter Qubits and Linear Optics." *Physical Review A* 71 (6): 060310.
- Bernien, H., B. Hensen, W. Pfaff, G. Koolstra, M. S. Blok, L. Robledo, T. H. Taminiau, et al. 2013. "Heralded Entanglement between Solid-State Qubits Separated by Three Metres." *Nature* 497 (7447): 86–90.
- Bodenstedt, S., I. Jakobi, J. Michl, I. Gerhardt, P. Neumann, and J. Wrachtrup. 2018. "Nanoscale Spin Manipulation with Pulsed Magnetic Gradient Fields from a Hard Disc Drive Writer." *Nano Letters* 18 (9): 5389–95.
- Bradac, Carlo, Weibo Gao, Jacopo Forneris, Matthew E. Trusheim, and Igor Aharonovich. 2019. "Quantum Nanophotonics with Group IV Defects in Diamond." *Nature Communications* 10 (1): 5625.
- Chakravarthi, Srivatsa, Christian Pederson, Zeeshawn Kazi, Andrew Ivanov, and Kai-Mei C. Fu. 2021. "Impact of Surface and Laser-Induced Noise on the Spectral Stability of Implanted Nitrogen-Vacancy Centers in Diamond." *Physical Review B, Condensed Matter* 104 (8): 085425.
- Corna, Andrea, Léo Bourdet, Romain Maurand, Alessandro Crippa, Dharmraj Kotekar-Patil, Heorhii Bohuslavskiy, Romain Laviéville, et al. 2018. "Electrically Driven Electron Spin Resonance Mediated by Spin–valley–orbit Coupling in a Silicon Quantum Dot." *Npj Quantum Information* 4 (1): 1–7.
- Degen, M. J., S. J. H. Loenen, H. P. Bartling, C. E. Bradley, A. L. Meinsma, M. Markham, D. J. Twitchen, and T. H. Taminiau. 2021. "Entanglement of Dark Electron-Nuclear Spin Defects in Diamond." *Nature Communications* 12 (1): 3470.
- De Santis, Lorenzo, Matthew E. Trusheim, Kevin C. Chen, and Dirk R. Englund. 2021. "Investigation of the Stark Effect on a Centrosymmetric Quantum Emitter in Diamond." *Physical Review Letters* 127 (14): 147402.
- Falk, Abram L., Paul V. Klimov, Bob B. Buckley, Viktor Ivády, Igor A. Abrikosov, Greg Calusine, William F. Koehl, Adám Gali, and David D. Awschalom. 2014. "Electrically and Mechanically Tunable Electron Spins in Silicon Carbide Color Centers." *Physical Review Letters* 112 (18): 187601.
- Faraon, Andrei, Charles Santori, Zhihong Huang, Victor M. Acosta, and Raymond G. Beausoleil. 2012. "Coupling of Nitrogen-Vacancy Centers to Photonic Crystal Cavities in Monocrystalline Diamond." *Physical Review Letters*.
<https://doi.org/10.1103/physrevlett.109.033604>.
- Gräupner, P., J. C. Pommier, A. Cachard, and J. L. Coutaz. 1992. "Electro-optical Effect in Aluminum Nitride Waveguides." *Journal of Applied Physics* 71 (9): 4136–39.
- Jakobi, Ingmar, Philipp Neumann, Ya Wang, Durga Bhaktavatsala Rao Dasari, Fadi El Hallak, Muhammad Asif Bashir, Matthew Markham, Andrew Edmonds, Daniel Twitchen, and Jörg Wrachtrup. 2017. "Measuring Broadband Magnetic Fields on the Nanoscale Using a Hybrid Quantum Register." *Nature Nanotechnology* 12 (1): 67–72.
- Kimiaee Asadi, F., S. C. Wein, and C. Simon. 2020. "Cavity-Assisted Controlled Phase-Flip Gates." *Physical Review A* 102 (1): 013703.
- Klimov, P. V., A. L. Falk, B. B. Buckley, and D. D. Awschalom. 2014. "Electrically Driven Spin Resonance in Silicon Carbide Color Centers." *Physical Review Letters* 112 (8): 087601.
- Koehl, William F., Bob B. Buckley, F. Joseph Heremans, Greg Calusine, and David D. Awschalom. 2011. "Room Temperature Coherent Control of Defect Spin Qubits in Silicon Carbide." *Nature* 479 (7371): 84–87.
- Kondo, Shinya, Reijiro Shimura, Takashi Teranishi, Akira Kishimoto, Takanori Nagasaki, Hiroshi Funakubo, and Tomoaki Yamada. 2021. "Linear Electro-Optic Effect in Ferroelectric HfO₂-Based Epitaxial Thin Films." *Japanese Journal of Applied Physics* 60 (7): 070905.

- Lee, Yuan, Eric Bersin, Axel Dahlberg, Stephanie Wehner, and Dirk Englund. 2020. "A Quantum Router Architecture for High-Fidelity Entanglement Flows in Quantum Networks." *arXiv [quant-Ph]*. arXiv. <http://arxiv.org/abs/2005.01852>.
- Maity, Smarak, Linbo Shao, Stefan Bogdanović, Srujan Meesala, Young-Ik Sohn, Neil Sinclair, Benjamin Pingault, et al. 2020. "Coherent Acoustic Control of a Single Silicon Vacancy Spin in Diamond." *Nature Communications* 11 (1): 193.
- Orphal-Kobin, Laura, Kilian Unterguggenberger, Tommaso Pregnolato, Natalia Kemf, Matthias Matalla, Ralph-Stephan Unger, Ina Ostermay, Gregor Pieplow, and Tim Schröder. 2022. "Optically Coherent Nitrogen-Vacancy Defect Centers in Diamond Nanostructures." *arXiv [quant-Ph]*. arXiv. <http://arxiv.org/abs/2203.05605>.
- Pfaff, Wolfgang, Tim H. Taminiau, Lucio Robledo, Hannes Bernien, Matthew Markham, Daniel J. Twitchen, and Ronald Hanson. 2012. "Demonstration of Entanglement-by-Measurement of Solid-State Qubits." *Nature Physics* 9 (1): 29–33.
- Pirot-Ladrière, M., T. Obata, Y. Tokura, Y-S Shin, T. Kubo, K. Yoshida, T. Taniyama, and S. Tarucha. 2008. "Electrically Driven Single-Electron Spin Resonance in a Slanting Zeeman Field." *Nature Physics* 4 (10): 776–79.
- Rodgers, Lila V. H., Lillian B. Hughes, Mouzhe Xie, Peter C. Maurer, Shimon Kolkowitz, Ania C. Bleszynski Jayich, and Nathalie P. de Leon. 2021. "Materials Challenges for Quantum Technologies Based on Color Centers in Diamond." *MRS Bulletin / Materials Research Society* 46 (7): 623–33.
- Rozpędek, Filip, Raja Yehia, Kenneth Goodenough, Maximilian Ruf, Peter C. Humphreys, Ronald Hanson, Stephanie Wehner, and David Elkouss. 2019. "Near-Term Quantum-Repeater Experiments with Nitrogen-Vacancy Centers: Overcoming the Limitations of Direct Transmission." *Physical Review. A* 99 (5): 052330.
- Ruf, M., M. J. Weaver, S. B. van Dam, and R. Hanson. 2021. "Resonant Excitation and Purcell Enhancement of Coherent Nitrogen-Vacancy Centers Coupled to a Fabry-Perot Microcavity." *Physical Review Applied* 15 (2): 024049.
- Sekiguchi, Yuhei, Kazuki Matsushita, Yoshiki Kawasaki, and Hideo Kosaka. 2022. "Optically Addressable Universal Holonomic Quantum Gates on Diamond Spins." <https://www.researchsquare.com/article/rs-1461563/latest.pdf>.
- Sekiguchi, Yuhei, Satsuki Okumura, and Hideo Kosaka. 2022. "Backward Propagating Quantum Repeater Protocol with Multiple Quantum Memories." *arXiv [quant-Ph]*. arXiv. <http://arxiv.org/abs/2205.04243>.
- Sipahigil, A., R. E. Evans, D. D. Sukachev, M. J. Burek, J. Borregaard, M. K. Bhaskar, C. T. Nguyen, et al. 2016. "An Integrated Diamond Nanophotonics Platform for Quantum-Optical Networks." *Science* 354 (6314): 847–50.
- Udvarhelyi, Péter, V. O. Shkolnikov, Adam Gali, Guido Burkard, and András Pályi. 2018. "Spin-Strain Interaction in Nitrogen-Vacancy Centers in Diamond." *Physical Review. B, Condensed Matter* 98 (7): 075201.
- Welte, Stephan, Bastian Hacker, Severin Daiss, Stephan Ritter, and Gerhard Rempe. 2017. "Cavity Carving of Atomic Bell States." *Physical Review Letters*. <https://doi.org/10.1103/physrevlett.118.210503>.

REVIEWERS' COMMENTS

Reviewer #1 (Remarks to the Author):

I think the new work relatd to strain driving with their platform addresses my core criticism of the impact and applicability of the work for suitability in a journal with broad audience such as Nature Comms. I am happy to recommend publication.

Reviewer #2 (Remarks to the Author):

My comments have been addressed properly. I'd like to recommend the publication now.

Reviewer #3 (Remarks to the Author):

I think the authors have addressed all the comments I made before and it should be almost ready for publication. However, I still want the authors to check all the coordinates in the figures since the horizontal axis looks inconsistent between the main text noting z and the supplementary material noting x for the length direction.